# Decoding Clozapine-Induced Agranulocytosis: Unraveling Interactions and Mitigation Strategies

**DOI:** 10.3390/pharmacy12030092

**Published:** 2024-06-12

**Authors:** Ali Alalawi, Enas Albalawi, Abdullah Aljohani, Abdullah Almutairi, Abdulraouf Alrehili, Areej Albalawi, Ahmed Aldhafiri

**Affiliations:** 1Drug Information Centre, Specialized Psychiatric Hospital in King Salman Bin Abdulaziz Medical City, Medina 42319, Saudi Arabia; adalalawi2@moh.gov.sa; 2Pharmacy Department, Specialized Psychiatric Hospital in King Salman Bin Abdulaziz Medical City, Medina 42319, Saudi Arabia; enassa@moh.gov.sa; 3Medication Safety, Specialized Psychiatric Hospital in King Salman Bin Abdulaziz Medical City, Medina 42319, Saudi Arabia; aaljohani106@moh.gov.sa; 4Inpatient Pharmacy, Specialized Psychiatric Hospital in King Salman Bin Abdulaziz Medical City, Medina 42319, Saudi Arabia; akalmutairi6@moh.gov.sa; 5Outpatient Pharmacy, Specialized Psychiatric Hospital in King Salman Bin Abdulaziz Medical City, Medina 42319, Saudi Arabia; abdulraoufa@moh.gov.sa; 6Laboratory Department, Madina Hospital in King Salman Bin Abdulaziz Medical City, Medina 42319, Saudi Arabia; areej-b2009@hotmail.com; 7Pharmacology and Toxicology Department, College of Pharmacy, Taibah University, Medina 42353, Saudi Arabia

**Keywords:** agranulocytosis, clozapine-induced agranulocytosis, refractory schizophrenia

## Abstract

Agranulocytosis represents a severe complication associated with the administration of clozapine. Clozapine is an antipsychotic medication that has demonstrated substantial efficacy in remediating refractory schizophrenia and various other psychiatric disorders. Nonetheless, it is crucial to monitor patients for neutropenia regularly during clozapine therapy. Therefore, this article aimed to delve into the prevalence of agranulocytosis during clozapine treatment by scrutinizing the extant literature to discern trends and correlations. This review endeavored to explore factors such as drug interactions, dose-related factors, duration of treatment, and genetic predispositions that could potentially influence the likelihood of patients developing agranulocytosis while undergoing clozapine therapy. Moreover, this review enunciates the ramifications of agranulocytosis on both patients and healthcare providers and meticulously evaluates the strategies to mitigate this risk and ensure optimal patient outcomes.

## 1. Introduction

Agranulocytosis is a critical medical condition, manifesting as an idiosyncratic reaction characterized by a profound reduction in the number of circulating neutrophils, resulting in heightened susceptibility to severe infections and potential mortality. This condition is a rare adverse effect that could be associated with clozapine therapy, presenting with an estimated incidence of 0.4% [1]. In hematological monitoring for clozapine treatment, it is essential to distinguish between neutropenia and agranulocytosis to ensure patient safety and manage side effects effectively. Neutropenia, with an estimated incidence of 3.8% during clozapine treatment, is defined as a reduction in neutrophil count below 1.5 × 10^9^/L, occasionally dropping below 2 × 10^9^/L [2,3]. This condition is often clinically silent and benign but may precede more severe declines in neutrophil levels [3]. In contrast, agranulocytosis is a severe condition characterized by neutrophil counts dropping below 0.5 × 10^9^/L, requiring urgent medical intervention [3]. Despite prior research conducted by Stubner et al. [4], who documented a notably elevated occurrence of neutropenia in clozapine-treated patients as opposed to those treated with haloperidol, risperidone, or promethazine [5], a recent study by Johanssen C-F et al. found no significant differences in neutropenia rates between patients exposed to clozapine and those who were not [6].

The mortality rate for patients treated with clozapine is 6.7 per 1000 patient-years, with a 95% confidence interval of 5.4 to 7.9, as indicated in 24 studies covering 217,691 patient-years [7]. Elderly patients have been observed to exhibit greater susceptibility to agranulocytosis, with an elevated risk of fatality [8]. Studies indicated that in patients aged 65 and older, 7% had developed agranulocytosis [9]. The risk of agranulocytosis is slightly elevated in females [10], while individuals of Asian descent display a 2.4-fold higher risk compared to those of Caucasian descent [11]. In relation to clozapine-induced neutropenia (CIN), Black African Caribbean patients face a 77% increased risk when compared to Caucasian individuals [11].

Clozapine, approved by the Food and Drug Administration (FDA) in 1989, has consistently demonstrated superior efficacy compared to alternative medications in remediating treatment-resistant schizophrenia (TRS). In 2002, it was exclusively prescribed as the sole therapeutic for addressing suicidality [12]. Meltzer and Okayli revealed that clozapine evidently reduced suicide attempts by up to 88% [13]. Additionally, clozapine demonstrated effectiveness in managing aggression and violence in patients as well as alleviating symptoms for those experiencing tardive dyskinesia as a side effect of other medications [12,14]. Furthermore, clozapine treatment is often considered cost-effective due to the reduction in hospitalization expenses and emergency care. A study by Hayhurst et al. [15] evidenced that clozapine therapy reduces the direct expenditure associated with treating TRS patients by approximately 7300 pounds sterling per patient, particularly in terms of hospitalization charges over a span of two years. Among various conventional antipsychotics, clozapine stands out as the optimal choice for addressing TRS [16].

Nevertheless, owing to its potential severity, it is imperative to investigate the underlying factors that contribute to the development of the CIA. Comprehending these risk factors is vital for making informed decisions regarding clozapine therapy, as remedies for the prevention of agranulocytosis remain elusive. Therefore, the rationale for conducting this review on clozapine-induced agranulocytosis is to offer an updated and comprehensive synthesis of recent research findings. This article reviews the extant literature to elucidate potential drug interactions and risk factors contributing to the development of CIA and delineate current management strategies.

## 2. Potential Mechanism Underlying CIA

Clozapine’s mechanism of action (MOA) is complex, with a broad impact on neurotransmitter systems, setting it apart from other antipsychotics. It exhibits a relatively lower affinity for D2 receptors, which reduces the incidence of extrapyramidal side effects compared to typical antipsychotics [17]. Its strong antagonism of serotonin 5-HT2A receptors may also contribute to treating schizophrenia’s negative cognitive symptoms and further reducing extrapyramidal risk [17,18]. Clozapine also interacts with adrenergic, cholinergic, and histaminergic receptors, influencing both therapeutic outcomes and side effects like hypersalivation and constipation [18]. Metabolic side effects, such as weight gain and diabetes, may result from its extensive receptor activity, impacting glucose and lipid metabolism [17]. Unique to clozapine is its potential to reverse dopamine supersensitivity, which is crucial for treatment-resistant cases [19].

The exact MOA responsible for the CIA is not yet fully understood. However, various ideas have been proposed to explain this occurrence, drawing on information from research discoveries. Clozapine is metabolized in the liver to stable metabolites like dimethyl-clozapine and clozapine N-oxide through the action of cytochrome P450 (CYP) enzymes (Figure 1) [20]. According to Pirmohammed and Park [20], it can also go through bioactivation to create a chemically reactive nitrenium ion, which can bind to glutathione to create glutathione conjugates like C6-glutathionyl-clozapine and C9-glutathionyl-clozapine. The reactive nitrenium ion, toxic to neutrophils, may contribute to the development of agranulocytosis by accelerating neutrophil apoptosis, potentially through the upregulation of pro-apoptotic proteins [21]. Sernoskie et al. [22] proposed that the metabolism of clozapine catalyzed by myeloperoxidase involves the conversion of the drug into a nitrenium ion, which is a highly reactive species. This, in turn, affects granulocytes and their precursors and ultimately affects apoptosis. Pirmohammed and Park [20] have suggested that CIA may involve an immune-mediated mechanism, indicated by antibodies against myeloperoxidase in patients’ serum and specific human leukocyte antigen (HLA) type associations. It is thought that oxidized clozapine metabolites bind to HLAs, forming haptens that stimulate an adaptive immune response through T-helper cells [22]. However, the evidence is not conclusive. Researchers have found that the metabolite N-desmethyl clozapine is more toxic to myeloid precursor cells than clozapine, which may lead to neutropenia and agranulocytosis [21]. In patients with CIA, a depletion of myeloid progenitor cells from the promyelocyte stage onward is often noted in the bone marrow [22]. Although N-desmethyl clozapine is highly toxic to hematopoietic progenitors, the toxicity observed in vitro requires concentrations exceeding those typically found in patient serum, and there is no established direct correlation between plasma levels of clozapine, its metabolite, and the onset of agranulocytosis [20].

Genetic factors may assume an additional role in the development of the CIA. Until recently, the majority of genetic association studies focused on candidate genes from the HLA system associated with the immune response. Certain HLA alleles, like HLA-DQB1*05:02 in Europeans and HLA-B*59:01 in the Japanese, may heighten the susceptibility to CIA by altering immune responses or increasing immune-mediated cytotoxicity [23]. Another study showed that the HLA-DQB1 6672G>C (rs113332494) variant significantly increased the risk of developing neutropenia and agranulocytosis in individuals of European ancestry, with a notably higher odds ratio for agranulocytosis than for neutropenia [24]. A genome-wide association study (GWAS) meta-analysis found a link between CIA and rs149104283, impacting the hepatic transporter genes SLCO1B3 and SLCO1B7, suggesting a pharmacokinetic mechanism [25]. Additionally, the Duffy-null genotype, a variant in the ACKR1 gene, is associated with lower neutrophil counts in African individuals and can lead to benign ethnic neutropenia, often mistaken for drug-induced neutropenia [23]. Recognizing genetic markers like the Duffy-null genotype could refine clozapine safety monitoring, particularly for those of African descent. The repeated mention of the role of genetic factors in the development of CIA and the emphasis on HLA alleles underscore the significance of these elements in comprehending the associated risk and mechanism underlying CIA in clozapine-treated patients [23]. Future research is crucial to clarifying these genetic influences and their functional importance, with diverse population studies essential due to varying genetic risks across ethnicities. Currently, there is a need for more extensive research to fully understand the genetic landscape of CIA and CIN.

## 3. Impact of Concurrent Medications on the Risk of Agranulocytosis in Clozapine-Treated Patients

The growing body of research specifically highlights the critical nature of monitoring drug interactions to mitigate the risks of adverse reactions, underscoring the necessity for heightened vigilance in pharmacovigilance practices. Despite the efficacy of clozapine, concerns surrounding potential drug interactions have led to a careful examination of the mechanisms behind these interactions and their impact on treatment outcomes. A comprehensive elucidation of these interactions will not only highlight the importance of constant vigilance among healthcare professionals but also serve as a reminder that a patient-centric approach is crucial for ensuring optimal treatment outcomes. Considering that only 40% of individuals with TRS respond positively to clozapine trials [26], it may be advantageous to develop effective clozapine augmentation strategies [27]. Clozapine can be combined with other psychotropic drugs, such as antidepressants, antipsychotics, and mood stabilizers [28,29]. The most promising approaches for augmenting clozapine include aripiprazole, fluoxetine, and sodium valproate. However, combining or augmenting clozapine with a second drug poses an increased risk of drug interactions, which may affect its effectiveness, produce side effects, or alter the mechanisms of action of one or both drugs [29]. When clozapine is combined with drugs that affect CYP enzymes, particularly CYP1A2, CYP3A4, and, to a lesser extent, CYP2D6, close monitoring of plasma concentrations is recommended to minimize the risk of dose-related toxicity [8].

Antiepileptic drugs (AEDs), particularly carbamazepine, have been empirically proven to interact with clozapine. Carbamazepine, which has a neutropenia risk of approximately 0.5% and an agranulocytosis risk of around 0.14%, is one of the leading drug-related causes of neutropenia and agranulocytosis [4,30]. Prior studies have revealed that patients taking clozapine and carbamazepine concurrently have an increased risk of developing agranulocytosis [31]. The exact mechanism of this interaction is not fully understood, but it has been hypothesized that carbamazepine, when used as a monotherapy, can decrease the ability of the bone marrow to produce white blood cells (WBCs) [31]. In addition to the increased risk of neutropenia from this drug combination, there is also a decrease in clozapine serum concentration by about 50% due to its potent induction of CYP3A4 and possibly CYP1A2 [32]. Therefore, patients treated with clozapine should be closely monitored for signs of agranulocytosis if they are also receiving carbamazepine, and alternative AEDs should be considered.

Another class of medications that can interact with clozapine and potentially induce agranulocytosis is the concurrent use of valproate and clozapine. Valproate, another antiepileptic drug, is widely used as a mood stabilizer in patients with bipolar disorder and is associated with an increased risk of agranulocytosis when used in combination with clozapine [33,34]. Yang et al. [35] conducted a retrospective study at the Taoyuan Psychiatric Center from 2006 to 2017, involving 1084 patients. The study documented blood dyscrasias induced by clozapine and valproate. In the group of 700 patients treated with clozapine, 2.9% developed neutropenia. Of the 319 patients receiving clozapine and valproate, 8.8% developed neutropenia. Yang et al. [35] provided evidence that the simultaneous use of valproate and clozapine is strongly linked to a three-fold increase in the risk of neutropenia (OR = 3.49, *p* < 0.001). The effect of sodium valproate on the risk of neutropenia was discovered to not be dose-dependent. An analysis of the combination of drugs used with valproate reported varying incidences of neutropenia: 44% in the combination group, 26% in the valproate monotherapy group, and 6% in the quetiapine monotherapy group [36]. The precise mechanism underlying this interaction remains vague to comprehension; however, it is believed that the effect of valproate on neutrophil production in the bone marrow may be responsible [33]. Valproate intensifies neutrophilic oxidative stress and apoptosis while also inhibiting the bone marrow’s production of granulocytes due to granulocyte-macrophage colony-stimulating factor (GM-CSF) suppression [33]. According to Mijovic and MacCabe [37], medications should be reassessed, and those that might induce neutropenia, namely sodium valproate and carbamazepine, should be discontinued. Instead, replacements with drugs that have minimal or no impact on neutrophils, such as levetiracetam, are advisable. Therefore, caution must be exercised when prescribing these medications together, and regular monitoring of the patient’s blood cell count is necessary to identify potential issues promptly. In cases where the absolute neutrophil count (ANC) starts to decline, clinicians should consider early withdrawal of valproate. This step is crucial to prevent further decline in ANC and avoid the subsequent discontinuation of clozapine. Mediations most commonly known to induce neutropenia and/or agranulocytosis are summarized in Table 1.

Antithyroid medications are also associated with an increased risk of agranulocytosis when used alongside clozapine [41]. Antithyroid medications are commonly used for the treatment of hyperthyroidism caused by overproduction of hormones. However, studies have depicted that the mechanism of certain antithyroid medications, such as propylthiouracil, appears to be associated with immunity [41]. Approximately 0.55% of patients experienced agranulocytosis due to propylthiouracil, while methimazole had a slightly lower incidence rate of 0.31% [38]. Consequently, patients taking clozapine should be closely monitored for signs of antithyroid-mediated agranulocytosis when they require thyroid inhibitor medications. Alternative thyroid management strategies should be considered to minimize this risk.

In addition to the aforementioned medications, the concomitant use of clozapine with other medications known to induce or worsen agranulocytosis should be avoided. Some of these medications include chemotherapeutic agents such as cyclophosphamide and docetaxel, as well as certain antibiotics like trimethoprim-sulfamethoxazole [42,43]. Chemotherapeutic agents and clozapine have the potential to independently cause agranulocytosis, and their concurrent use could exacerbate the risk of this severe hematological adverse event. Chemotherapeutic agents are known to cause neutropenia owing to their cytotoxic effects on the bone marrow, where active cell division occurs, thus depleting hematopoietic stem cells and leading to conditions such as agranulocytosis, neutropenic sepsis, and febrile neutropenia [42]. Neutropenia can be predictable and dose-related, as is evident in cytotoxic chemotherapy, and the risk of infection depends on the adequacy of the bone marrow reserve [44]. The use of clozapine along with these medications greatly enhances the risk of developing agranulocytosis; therefore, alternative treatment options should be considered for patients who require them.

## 4. Impact of Comorbid Conditions on the Risk of Agranulocytosis in Clozapine-Treated Patients

Although the exact mechanism underlying CIA is not fully understood, it is thought to involve immunological factors. Patients with preexisting hematological disorders, such as leukopenia or myeloproliferative disorders, may contribute to the occurrence of agranulocytosis [45]. Consequently, it is recommended not to start clozapine therapy if the patient has a low baseline WBC or neutrophil count, a history of myeloproliferative disorders, or previous CIA [46]. These conditions weaken the immune system and increase the susceptibility to infections when combined with clozapine. In contrast, some evidence suggests that leukocytosis from clozapine treatment may be harmless, questioning whether preexisting leukopenia exacerbates the risk of agranulocytosis [47]. Chen et al. [48] emphasized the role of genetic factors in managing clozapine treatment in individuals with preexisting disorders.

Various studies and clinical observations have derived insights into the influence of comorbid conditions on the risk of agranulocytosis in clozapine-treated patients. A case study highlighted delayed CIA in a patient with multiple sclerosis, emphasizing the importance of extended blood count monitoring beyond the initial six months of clozapine treatment [49]. Clozapine-treated patients with cerebrovascular, cardiovascular, diabetes, or multiple recent respiratory infections requiring antibiotics are at increased risk for severe COVID-19 complications [50]. Gee and Taylor [51] found that COVID-19 infection can cause mild neutropenia in patients established on clozapine for over six months. Based on genetic profiles, pharmacogenomic factors can guide the intensity of hematological monitoring [52].

In summary, the risk of agranulocytosis in patients treated with clozapine is multifaceted, especially in patients with comorbid conditions. Continuous hematological monitoring, understanding of genetic influences, and considering comorbidities are vital for effective and safe clozapine treatment.

## 5. Effect of Therapeutic Duration and Drug Dosage on the Induction of Clozapine-Mediated Agranulocytosis

Debate on whether the duration and dosage of clozapine therapy influence the risk of agranulocytosis is ongoing. This study explored whether longer treatment durations and higher doses of clozapine were correlated with an increased risk of agranulocytosis. We examined various studies that have focused on the relationship between clozapine therapy duration and the risk of agranulocytosis.

Several studies have investigated the relationship between the duration of clozapine therapy and the risk of agranulocytosis. According to Myles et al. [2], the likelihood of developing agranulocytosis is elevated within the initial 3–6 months of treatment, although neutropenia can manifest in any given period. This suggests that the risk of agranulocytosis may be higher during the initial stages of clozapine treatment. Furthermore, another study by Andersohn et al. [53] found that CIA typically occurs a few weeks after treatment initiation. Their findings indicated a median interval of 56 days and an average period with agranulocytosis of 12 days, further supporting the association between treatment duration and the development of agranulocytosis.

In addition to treatment duration, there is no evidence to suggest that high doses of clozapine increase the risk of agranulocytosis. Some research has suggested that the therapeutic serum concentration of clozapine typically ranges from 350 to 420 ng/mL [32]. Concentrations above 1000 ng/mL are not recommended due to the increased risk of toxicity [32]. It is important for clinicians to monitor serum levels to balance efficacy and safety, especially when clozapine is used in combination with other drugs that can affect its metabolism. Notably, many side effects are associated with higher clozapine levels, although some are not related to serum concentration. Bishara and Taylor [8] stated that there is no link between higher doses or plasma concentrations of clozapine and the occurrence of agranulocytosis. Nielsen et al. [5] examined the medical reasons for discontinuing clozapine and concluded that certain severe side effects, such as agranulocytosis and myocarditis, were idiosyncratic in nature and not dose-dependent. In contrast, other notable side effects, such as tachycardia and seizures, appeared to be dose-dependent.

However, it is important to note that the relationship between treatment duration, high dose, and agranulocytosis is not entirely straightforward. Some studies failed to find a considerable association between treatment duration and the risk of agranulocytosis. For example, Dunk et al. [54] suggested that although the likelihood of agranulocytosis decreases over time, there have been instances of agranulocytosis occurring after prolonged periods of treatment.

The conflicting findings regarding the influence of treatment duration and high doses of clozapine on agranulocytosis should be interpreted with caution. Other factors, such as genetic predisposition, age, and concomitant medication use, may contribute to the development of agranulocytosis. Furthermore, differences in the study design, sample size, and participant characteristics may account for the varying results.

## 6. Management and Prevention of Agranulocytosis during Clozapine Therapy

As the risk of agranulocytosis can be critical during the course of clozapine therapy, proper management strategies are imperative to prevent this condition. This review explores various methods for managing and preventing agranulocytosis during clozapine therapy, supported by reputable sources.

Regular monitoring of the WBC count is crucial for managing the CIA. According to the Maudsley prescribing guidelines [55], monitoring the ANC, which measures the number of circulating neutrophils, is key to the early identification and prevention of agranulocytosis. Regular ANC monitoring allows healthcare professionals to detect any rapid or substantial drop in the neutrophil count. They also suggested that increasing the monitoring frequency during the first few months of therapy could further reduce the risk of agranulocytosis.

Proper patient education and understanding of the risks associated with clozapine therapy are other essential components for managing and preventing agranulocytosis. Sedhai et al. [45] emphasized the importance of educating patients about the signs and symptoms of agranulocytosis, such as fever, sore throat, and flu-like symptoms, to encourage prompt reporting to healthcare providers. Early recognition of these symptoms can lead to timely intervention and minimize the severity of agranulocytosis.

In addition to regular monitoring and patient education, the implementation of a comprehensive risk management program is crucial for preventing agranulocytosis during clozapine therapy. Risk management programs should involve strict adherence to guidelines and protocols established by regulatory agencies and professional bodies. The FDA and the European Agency for the Evaluation of Medicinal Products (EMEA) provide specific guidelines for monitoring and managing agranulocytosis during clozapine therapy [56,57]. These guidelines include recommendations for baseline and regular blood tests, dose adjustments, and discontinuation protocols, if necessary.

Furthermore, some studies have explored the potential role of genetic testing in identifying patients at higher risk of agranulocytosis due to clozapine therapy. A review by Shad [58] suggested that certain genetic markers may provide valuable insights into an individual’s susceptibility to agranulocytosis. These markers could potentially help identify patients at higher risk and require closer monitoring or alternative treatment options. The research emphasizes the role of specific HLA alleles and other genetic polymorphisms in the susceptibility to CIA [58]. Genetic factors play a role, with certain haplotypes such as HLA-B38, DRB1*0402, DRB4*0101, and DQB1*0302 linked to agranulocytosis in Ashkenazi Jews [59]. Additionally, the DQB1 gene’s single nucleotide polymorphism (6672G>C) considerably raises the odds of agranulocytosis [60]. Although genetic tests hold significant potential, they have not yet reached a level of clinical utility, necessitating further research. Currently, no established guidelines exist to direct their clinical application.

Finally, it is important to consider alternative treatment options for patients who are at a higher risk or have a documented history of agranulocytosis with clozapine. Although clozapine is considered the gold standard treatment for TRS, other antipsychotic medications may be suitable alternatives in specific cases. The British Journal of Medical Practitioners suggested that combinations of antipsychotics such as olanzapine and amisulpride might be effective in cases where clozapine is unsuitable [61]. Additionally, various augmentation strategies, including other antipsychotics, mood stabilizers, benzodiazepines, lithium, electroconvulsive therapy, and repetitive transcranial magnetic stimulation, have been used, although there is a lack of strong evidence for these interventions [61]. It is also noted that combining depot antipsychotics with oral drugs from different classes is a generally accepted practice [61]. A systematic review of the use of granulocyte colony-stimulating factor in clozapine rechallenges by [62] found that patients who had previously experienced CIA were treated with granulocyte colony-stimulating factor (G-CSF), such that, for example, filgrastim might shorten the period of agranulocytosis by approximately five days. In another study by Kanaan and Kerwin [63], it was found that out of 25 patients who underwent clozapine rechallenge with lithium co-prescription, only 1 patient (4%) experienced a second episode of neutropenia or agranulocytosis. Focosi et al. [64] have demonstrated that lithium administration raises G-CSF levels in urine and boosts G-CSF production by peripheral blood mononuclear cells. This suggests lithium’s primary in vivo role is to enhance granulopoiesis, the formation of granulocytes such as neutrophils [64]. Additionally, administering lithium to individuals with normal hematological profiles results in increased neutrophil production in the bone marrow and a subsequent rise in peripheral blood neutrophils due to enhanced G-CSF release [64]. In a review conducted by Boazak et al. [65], oral lithium was utilized to boost the production of leucocytes; however, this requires thorough monitoring. There are concerns, such as whether lithium might obscure an upcoming CIA or whether discontinuing lithium is safe after white cell count recovery.

## 7. Conclusions

This review adds to the existing literature by presenting updated data and findings, providing a comprehensive overview of clozapine-induced agranulocytosis, and offering practical recommendations for clinicians. The effective management of agranulocytosis in the context of clozapine therapy necessitates the implementation of various strategies, including routine monitoring of ANC, provision of comprehensive patient education, adherence to treatment guidelines, and genetic testing for susceptibility. Although clozapine is remarkably effective in remediating schizophrenia and mitigating suicidal ideation, it is crucial to be aware of its potential interactions with anticonvulsant drugs, namely carbamazepine and valproate, which may aggravate the risk of agranulocytosis. As the existing evidence does not provide conclusive support for the notion that extended periods of clozapine treatment or higher doses are associated with an elevated risk of agranulocytosis, this correlation remains ambiguous, thereby necessitating further investigation. It is imperative to present subsequent opportunities for developing preventive measures and alternative therapeutic strategies that can facilitate the mitigation of any associated side effects.

## Figures and Tables

**Figure 1 pharmacy-12-00092-f001:**
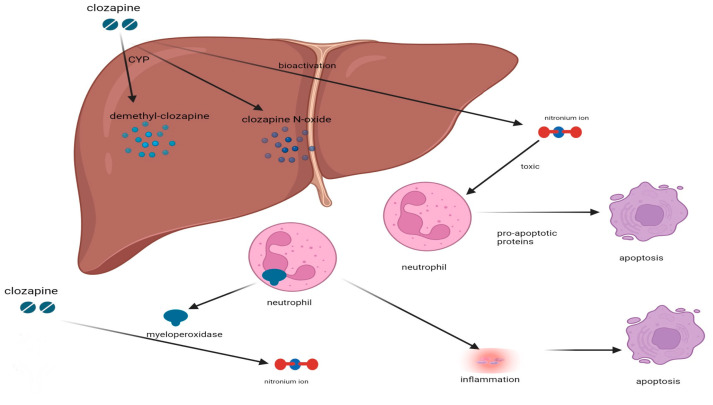
Clozapine is metabolized in the liver into stable metabolites like dimethyl-clozapine and clozapine N-oxide by cytochrome P450 enzymes and undergoes bioactivation to produce a reactive nitrenium ion. This ion, toxic to neutrophils, potentially accelerates their apoptosis and may lead to agranulocytosis, partly through the activation of pro-apoptotic proteins. The metabolism of clozapine by myeloperoxidase also results in this reactive ion, negatively promoting apoptosis.

**Table 1 pharmacy-12-00092-t001:** Incidence of agranulocytosis and neutropenia with medications.

Medications	Interaction Mechanism	Incidence of Agranulocytosis	Incidence of Neutropenia	Mitigation Strategy	References
Carbamazepine		0.14%	0.5%	Monitor blood levels; consider alternative AEDs	[4,30]
Clozapine		0.4%	2.9%	Monitor blood levels	[1,35]
Valproate with clozapine	Sodium valproate may potentiate CIN by inhibiting its metabolism, enhancing oxidative stress, and suppressing bone marrow function.		8.8%	Monitor blood levels; consider alternative AEDs	[33,35]
Antithyroid Medications (e.g., propylthiouracil, methimazole)		Propylthiouracil: 0.55%; Methimazole: 0.31%		Monitor blood levels; consider alternative Antithyroid Medications	[38]
Proton Pump Inhibitors Medications (e.g., omeprazole) with clozapine	PPIs induce CYP1A2, which increases N-desmethylclozapine levels and possibly influences flavin-containing monooxygenase (FM-O3), increasing nitrenium ion production.		Significantly higher when combined with PPIs.	Monitor blood levels; consider using PPIs with lower interaction potential, like rabeprazole; prefer alternative gastroesophageal reflux disease treatments over PPIs when possible, like misoprostol	[39]
Nirmatrelvir-Ritonavir (Paxlovid) with clozapine	Interaction likely involves inhibition of CYP1A2 by the ritonavir component, potentially increasing clozapine levels; concomitant SARS-CoV-2 infection may exacerbate neutropenia.		Increased risk of neutropenia in the context of SARS-CoV-2 infection and multiple drug interactions.	Monitor blood levels; consider adjusting the clozapine dosage	[40]

## Data Availability

No new data were created or analyzed in this study. Data sharing is not applicable to this article.

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
