# Peer review of "Decoding Clozapine-Induced Agranulocytosis: Unraveling Interactions and Mitigation Strategies"

_pharmacy, 2024, doi:10.3390/pharmacy12030092_

Round 1
Reviewer 1 Report
Comments and Suggestions for Authors
Please provide more explanation for the claim that treatment with clozapine is an expensive therapy. The price of clozapine is not expensive because it is off patent. Therefore, other explanation with reference to literature is needed. As area in need of further explanation is the interaction pf clozapine with ion channels.
Author Response
Point 1: Please provide more explanation for the claim that treatment with clozapine is an expensive therapy. The price of clozapine is not expensive because it is off patent. Therefore, other explanation with reference to literature is needed.
Response 1: Thank you for pointing this out. We agree with this comment. we removed the word "expensive" and clarified the meaning [Furthermore, clozapine treatment is often considered cost-effective due to the reduction in hospitalization expenses and emergency care]
Point 1: As area in need of further explanation is the interaction pf clozapine with ion channels.
Response 2: Thanks for your comment. We do agree with your point, but our research is specifically focused on agranulocytosis. Although the interaction of clozapine with ion channels does affect other serious conditions like myocarditis, these topics are outside our current study's scope but are important areas for future research.
Reviewer 2 Report
Comments and Suggestions for Authors
This paper aims at providing an overview of the state of the art on clozapine induced neutropenia and agranulocytosis. It is an interesting paper and easy to read for people who might not be too familiar with the subject.
It is quite long and some sections can be summarised more effectively.
It would be good to have a table/figure - The only table does not add much and the rationale of it is not clear.
Please find below a few more detailed reccommendations:
- Introduction: the authors should clearly differentiate between neutropenia and agranulocytosis, as the mechanism behind these phenomena may be different (Taylor D, Vallianatou K, Whiskey E, Dzahini O, MacCabe J. Distinctive pattern of neutrophil count change in clozapine-associated, life-threatening agranulocytosis. Schizophrenia (Heidelb). 2022 Mar 14;8(1):21. doi: 10.1038/s41537-022-00232-0. PMID: 35288577; PMCID: PMC8920060). It looks like there is confusion between the two concepts in the first paragraph
- Line 1: Agranulocytosis can also be non-idiosyncratic in other medical conditions, please re-phrase the first sentence
- Line 41-43: There have been studies with conflicting findings - please add a sentence on this (for example: Johannsen CF, Petersen TS, Nielsen J, Jørgensen A, Jimenez-Solem E, Fink-Jensen A. Clozapine- and non-clozapine-associated neutropenia in patients with schizophrenia: a retrospective cohort study. Ther Adv Psychopharmacol. 2022 Mar 5;12:20451253211072341. doi: 10.1177/20451253211072341. PMID: 35273789; PMCID: PMC8902187.)
- Line 58: please add reference for reduction in violence (e.g. Bhavsar V, Kosidou K, Widman L, Orsini N, Hodsoll J, Dalman C, MacCabe JH. Clozapine Treatment and Offending: A Within-Subject Study of Patients With Psychotic Disorders in Sweden. Schizophr Bull. 2020 Feb 26;46(2):303-310. doi: 10.1093/schbul/sbz055. PMID: 31150553; PMCID: PMC7442333)
- Line 60: can the authors clarify what they mean with “expensive”? Evidence, as rightly highlighted later, show the opposite
- Line 80: clozapine does not cause dry mouth, it causes hypersalivation - please correct.
- Lines 110-127 – please discuss findings of and add reference of: Konte, B., Walters, J.T.R., Rujescu, D. et al. HLA-DQB1 6672G>C (rs113332494) is associated with clozapine-induced neutropenia and agranulocytosis in individuals of European ancestry. Transl Psychiatry 11, 214 (2021). https://doi.org/10.1038/s41398-021-01322-w
- Line 135-136: sentence is unclear and possibly slightly redundant.
- Line 143-144: please provide reference for this recommendation - clozapine augmentation strategies have poor evidence base and this should be highlighted.
- Line 156: please remove the “++”
- Table 1 does not add much to the paper. It is not comprehensive (it only shows a few potential concomitant medications) and the choice of medications listed is not systematic nor the rationale is described
- Line 222: please add reference
- Lines 224-226: this concept has already been discussed – no need to repeat
- Lines 227-228: please add reference
- Reference 51 does not seem to correspond to what is referred to in the text
- 320-329: please highlight that although meaningful, genetic tests are not clinically useful yet and more research is needed – no guidelines exist yet.
- 340-343: this sentence is unclear
Comments on the Quality of English LanguageEnglish is overall good and comprehensible. However, a few sentences are unclear and should be re-written.
Author Response
Point 1: It would be good to have a table/figure
Response 1: Figure 1: Clozapine is metabolized in the liver into stable metabolites like dimethyl-clozapine and clozapine N-oxide by cytochrome P450 enzymes and undergoes bioactivation to produce a reactive nitrenium ion. This ion, toxic to neutrophils, potentially accelerates their apoptosis and may lead to agranulocytosis, partly through the activation of pro-apoptotic proteins. The metabolism of Clozapine by myeloperoxidase also results in this reactive ion, negatively promoting apoptosis.
Point 2: Introduction: the authors should clearly differentiate between neutropenia and agranulocytosis, as the mechanism behind these phenomena may be different (Taylor D, Vallianatou K, Whiskey E, Dzahini O, MacCabe J. Distinctive pattern of neutrophil count change in clozapine-associated, life-threatening agranulocytosis. Schizophrenia (Heidelb). 2022 Mar 14;8(1):21. doi: 10.1038/s41537-022-00232-0. PMID: 35288577; PMCID: PMC8920060). It looks like there is confusion between the two concepts in the first paragraph
Response 2: we clearly differentiate between neutropenia and agranulocytosis with the suggested reference In hematological monitoring for clozapine treatment, it is essential to distinguish between neutropenia and agranulocytosis to ensure patient safety and manage side effects effectively. Neutropenia, with an estimated incidence of 3.8% during clozapine treatment, is defined as a reduction in neutrophil count below 1.5 × 10^9/L, occasionally dropping below 2 × 10^9/L [2, 3]. This condition is often clinically silent and benign but may precede more severe declines in neutrophil levels [3]. In contrast, agranulocytosis is a severe condition characterized by neutrophil counts dropping below 0.5 × 10^9/L, requiring urgent medical intervention [3].
Point 3: Line 1: Agranulocytosis can also be non-idiosyncratic in other medical conditions, please re-phrase the first sentence
Response 3: This condition is a rare adverse effect could associated with clozapine therapy, presenting with an estimated incidence of 0.4% [1]
Point 4: Line 41-43: There have been studies with conflicting findings - please add a sentence on this (for example: Johannsen CF, Petersen TS, Nielsen J, Jørgensen A, Jimenez-Solem E, Fink-Jensen A. Clozapine- and non-clozapine-associated neutropenia in patients with schizophrenia: a retrospective cohort study. Ther Adv Psychopharmacol. 2022 Mar 5;12:20451253211072341. doi: 10.1177/20451253211072341. PMID: 35273789; PMCID: PMC8902187.)
Response 4: we added the sentence with suggested reference Despite prior research conducted by Stubner et al. [2] documented a notably elevated occurrence of neutropenia in clozapine-treated patients as opposed to those treated with haloperidol, risperidone, or promethazine [3], a recent study by Johanssen C-F et al. found no significant differences in neutropenia rates between patients exposed to clozapine and those who were not [4]
Point 5: Line 58: please add reference for reduction in violence (e.g. Bhavsar V, Kosidou K, Widman L, Orsini N, Hodsoll J, Dalman C, MacCabe JH. Clozapine Treatment and Offending: A Within-Subject Study of Patients With Psychotic Disorders in Sweden. Schizophr Bull. 2020 Feb 26;46(2):303-310. doi: 10.1093/schbul/sbz055. PMID: 31150553; PMCID: PMC7442333)
Response 5: we used the suggested reference [14]
Point 6: Line 60: can the authors clarify what they mean with “expensive”? Evidence, as rightly highlighted later, show the opposite
Response 6: Thank you for pointing this out. We agree with this comment. we removed the word "expensive" and clarified the meaning Furthermore, clozapine treatment is often considered cost-effective due to the reduction in hospitalization expenses and emergency care.
Point 7: Line 80: clozapine does not cause dry mouth, it causes hypersalivation - please correct.
Response 7: we did the corect word also interacts with adrenergic, cholinergic, and histaminergic receptors, influencing both therapeutic outcomes and side effects like hypersaliviation and constipation
Point 8: Lines 110-127 – please discuss findings of and add reference of: Konte, B., Walters, J.T.R., Rujescu, D. et al. HLA-DQB1 6672G>C (rs113332494) is associated with clozapine-induced neutropenia and agranulocytosis in individuals of European ancestry. Transl Psychiatry 11, 214 (2021). https://doi.org/10.1038/s41398-021-01322-w
Response 8: we added Another study showed that the HLA-DQB1 6672G>C (rs113332494) variant significantly increased the risk of developing neutropenia and agranulocytosis in individuals of European ancestry, with notably higher odds ratio for agranulocytosis than for neutropenia [22].
Point 9: Line 135-136: sentence is unclear and possibly slightly redundant.
Response 9: We removed the sentence and add the efficacy of clozapine. One drug that has drawn particular attention is clozapine, which has exhibited considerable success in managing symptoms; however, concerns over its safety due to drug interactions have risen within the medical community. Despite the efficacy of clozapine
Point 10: Line 143-144: please provide reference for this recommendation - clozapine augmentation strategies have poor evidence base and this should be highlighted
Response 10: we provided the reference, removed it is crucial and added it may be advantageous
Point 11: Line 156: please remove the “++”
Response 11: we removed the ++
Point 12: Table 1 does not add much to the paper. It is not comprehensive (it only shows a few potential concomitant medications) and the choice of medications listed is not systematic nor the rationale is described
Response 12: we deleted the table and the sentence Table 1 summarizes the incidence rates of neutropenia and/or agranulocytosis among medications commonly known to induce these conditions.
Point 13: Line 222: please add reference
Response 13: we deleted Comorbidities substantially influenced the risk of agranulocytosis in the patients receiving clozapine.
Point 14: Lines 224-226: this concept has already been discussed – no need to repeat
Response 14: we agree and deleted Neutrophils, which are essential for the immune response, are thought to be destroyed by the antibodies induced by clozapine metabolism [45], leading to increased vulnerability to infections.
Point 15: Lines 227-228: please add reference
Response 15: we provided the reference, removed have a higher risk and added may contribute to the occurance
Point 16: Reference 51 does not seem to correspond to what is referred to in the text
Response 16: we deleted A study from India on low-dose clozapine in patients with movement disorders found that CIA can occur even at lower doses [56]. They observed CIA in 0.8% of patients treated with low doses of clozapine.
Point 17: 320-329: please highlight that although meaningful, genetic tests are not clinically useful yet and more research is needed – no guidelines exist yet
Response 17: we agree and added Although genetic tests hold significant potential, they have not yet reached a level of clinical utility, necessitating further research. Currently, no established guidelines exist to direct their clinical application.
Point 18: 340-343: this sentence is unclear
Response 18: we deleted the last sentence to make it clear, and the average duration from the initiation of G-CSF to recovery was seven days.
Reponse 19: English language was revised by experts and grammarly.
Reviewer 3 Report
Comments and Suggestions for Authors
Thank you for the opportunity to review this outstanding manuscript reviewing clozapine-induced agranulocytosis (CIA). The potential mechanism of CIA, risk factors, including dose and duration of clozapine use, concomitant drugs, and comorbid conditions, and the treatment and prevention of CIA, are discussed in great detail. The manuscript appropriately emphasizes that much is still unkown regarding CIA, particularly the need for more studies to identify potential genetic profiles that increase risk. I agree with your conclusions, which are based solidly on the evidence-based information provided in the manuscript.
Author Response
Point 1: Thank you for the opportunity to review this outstanding manuscript reviewing clozapine-induced agranulocytosis (CIA). The potential mechanism of CIA, risk factors, including dose and duration of clozapine use, concomitant drugs, and comorbid conditions, and the treatment and prevention of CIA, are discussed in great detail. The manuscript appropriately emphasizes that much is still unkown regarding CIA, particularly the need for more studies to identify potential genetic profiles that increase risk. I agree with your conclusions, which are based solidly on the evidence-based information provided in the manuscript
Response 1: Thank you for your reviewing and agremments. Appreciated !
